# INFLUENTIAL LANGUAGE DATA SELECTION VIA GRADIENT TRAJECTORY PURSUIT

## ABSTRACT

Curating a desirable dataset for training has been the core of building highly capable large language models (Touvron et al., 2023; Achiam et al., 2023; Team et al., 2024). Gradient influence scores (Pruthi et al., 2020; Xia et al., 2024) have been shown to be correlated with model performance and are commonly used as the criterion for data selection. However, existing methods are built upon either individual sample rankings or inefficient matching process, leading to suboptimal performance or scaling up issues. In this paper, we propose *Gradient Trajectory Pursuit (GTP)*, an algorithm that performs pursuit of gradient trajectories via jointly selecting data points under an $L0$-norm regularized objective. The proposed algorithm highlights: (1) *joint selection* instead of independent top-$k$ selection, which automatically de-duplicates samples; (2) higher efficiency with compressive sampling processes, which can be further sped up using a distributed framework. In the experiments, we demonstrate the algorithm in both in-domain and target-domain selection benchmarks and show that it outperforms top-$k$ selection and competitive algorithms consistently, for example, our algorithm chooses as low as 0.5% data to achieve full performance on the targeted instruction tuning tasks.

## 1 INTRODUCTION

Large language models encompasses an enormous amount of knowledge acquired through pretraining corpus (Brown et al., 2020; Jiang et al., 2023; Touvron et al., 2023). A crucial component that makes language models practically useful is the post-training phase, which empower the model with a variety of extra abilities and skills, such as aligning language models to follow human instructions (Ouyang et al., 2022; Taori et al., 2023; Wang et al., 2023b; Xu et al., 2023), teaching models the tool use (Schick et al., 2023) or exploring environments (Carta et al., 2023; Lin et al., 2023).

Serving as the core of model training, choosing and utilizing desirable datasets are critical for both quickly adapting models given a fixed budget (Xie et al., 2023) and maximizing the performance for a targeted task (Wang et al., 2023a; Xia et al., 2024). An automatic data selection algorithm that chooses a subset based on certain information is gaining its needs given the massive amount of corpora available online. Gradient information, which reflects the optimization process and has direct correlation with the final model performance, is one of the most common criterion for data selection (Mirzasoleiman et al., 2020; Killamsetty et al., 2021; 2022; Yang et al., 2023; Pan et al., 2024; Xia et al., 2024). However, how to effectively utilize gradient for actual selection process is still a challenging problem. Top-$k$ selection is fast and the most straightforward way, where data quality is ranked via the similarity scores of individual gradient vectors and the main gradient trajectory (Xia et al., 2024). Nonetheless, due to its independence assumption, its performance is often suboptimal compared to joint selection (Evans et al., 2024). Orthogonal Matching Pursuit (Cai & Wang, 2011; Killamsetty et al., 2021) can perform joint selection through interatively removing subspace projections, but is highly time-inefficient and requires solving a non-negative least square for as many iterations as the target subset sample size.

In this paper, we introduce *Gradient Trajectory Pursuit (GTP)*, an algorithm that selects data samples through matching the trajectory on a gradient subspace. As discussed in previous works, a model's optimization process can be seen as the Langevin dynamic of an energy-based model (Welling & Teh, 2011), and matching the optimization process can lead to model weights with similar performance (Zhao et al., 2021; Cazenavette et al., 2022). This sets the base rationale for our algorithm. Further drawing inspiration from studies demonstrating that the optimization process happens in a subspace (Gur-Ari et al., 2018; Frankle & Carbin, 2018; Singhal et al., 2023), we project the gradients onto a small subspace and

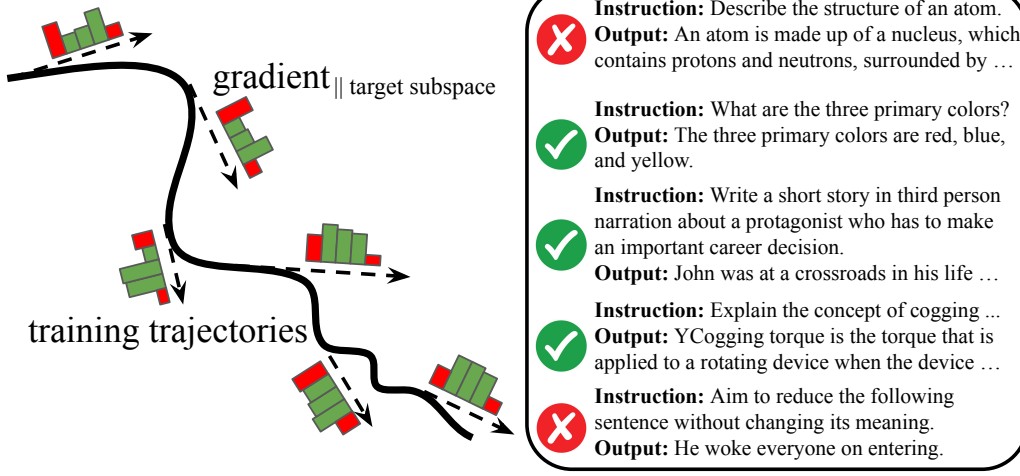

Figure 1: Our method selects a subset of training data through pursuit process of gradient trajectories on a target subspace during *warmup* training. The algorithm automatically de-duplicates samples instead of simply selecting the top-$k$ data. Text examples in the figure are drawn from instruction tuning datasets.

greatly reduces the memory cost (both on disk and RAM) during selection. This design also follows and is backed up by a recent work (Xia et al., 2024). The core of our algorithm is *joint data selection*, where data weights are solved through compressive sampling matching pursuit process. The pursuit process is equivalent to starting from a top-$k$ intialization, iteratively determines the proper combination of data samples and implicitly performs de-duplication. We also showcase a distributed version of the algorithm which further reduces the selection computation time, indicating its potential for scaling up to larger datasets.

**Our contributions are summarized as follows:**

- We propose an algorithm that jointly selects influential language data given a subset budget, via matching trajectories in a subspace. The algorithm is both effective on select compact dataset and up to 17x more efficient compared to a vanilla orthogonal matching pursuit-based algorithms and automatically performs de-duplication compared to top-$k$. Our algorithm also support distributed selection across machines.

- We experimentally show great improvements over the prior methods on two challenging benchmarks for language models, demonstrating our algorithm's usage on both in-domain and target-domain data selection. Specifically, we show that our algorithm can select only 0.5% of the whole dataset to achieve full performance on targeted instruction tuning.

- We in-depth analyzes our algorithms, including the convergence, the progression of subset quality over iterations, and comparison with top-$k$ selected data on ALFWorld agent dataset, providing understandings on how gradient similarities can lead to duplication and our algorithm's adjustment process for creating a compact subset.

## 2 RELATED WORKS

**Coreset data selection.** Building an effective training dataset has evolved to the cornerstone of foundation model construction. Due to the heavy cost of human annotation and filtering, a flux of works are developed to perform automated data selections. *Features of data* carry information that discriminates samples and are often used for selecting representative points (Sener & Savarese, 2017; Kaushal et al., 2019; Xia et al., 2020; Wang et al., 2020; Xia et al., 2022; Xie et al., 2023). N-gram features (Xie et al., 2023) are used for re-weighting and selecting samples for pre-training, demonstrating strong correlations with target domains. Deep learning features (Zhang et al., 2018; Hanawa et al., 2020; Xia et al., 2024) are common choices for effective baselines measuring data similarities. Although frequently adopted, feature-based methods are not guaranteed to have correlation with model optimization. *Gradient information* is another natural choice for selecting influential data (Yu et al., 2020; Mirzasoleiman et al., 2020; Mindermann et al., 2022; Han et al., 2023; Xia et al., 2024). Different from features, gradients often reflect more information about the

model optimization process (Mirzasoleiman et al., 2020; Xia et al., 2024), but often are used for in-domain selection (Mirzasoleiman et al., 2020; Killamsetty et al., 2021) and suffer from the curse of dimensionality, given the current model sizes (Brown et al., 2020; Jiang et al., 2023). Instead, our work allows for both in-domain selection and target domain transfer, and focuses low-dimensional representations. Besides the choices of representations, selection algorithms also play a crucial role. Various sampling methods or clustering algorithms are explored (Sener & Savarese, 2017; Kaushal et al., 2019; Xia et al., 2020; Wang et al., 2020; Mirzasoleiman et al., 2020; Killamsetty et al., 2021). *Data Models* (Ilyas et al., 2022; Engstrom et al., 2024), *LLM-asking* (Sachdeva et al., 2024), or information optimization (Everaert & Potts, 2023) have shown their effectiveness. Note that among these methods, gradient-based ones usually have more theoretical supports due to its connection with model optimization.

**Data influence.** Computing the data influence for model is another line of work for determining the importance of samples. Influence function for deep neural networks (Koh & Liang, 2017; Koh et al., 2019) can approximate the influence of dropping training samples. Trajectory influence (Pruthi et al., 2020; Han et al., 2023; Xia et al., 2024) is different definition introduced to measure how each data sample affects the whole model training dynamics. Top-$k$ selection and variants are adopted for selecting data with higher trajectory influences (Han et al., 2023; Xia et al., 2024).

**Model training dynamics and SDEs.** It has been shown that machine learning model trainings are also a diffusion processes (Welling & Teh, 2011; Hoffman et al., 2017; Wenzel et al., 2020) defined through stochastic differential equations. Commonly known in generative models, diffusion processes converge to desired distributions when following the target score vector (gradient) field (Song & Ermon, 2019; Song et al., 2020; Ho et al., 2020). Several works (De Bortoli, 2022; Zhu et al., 2024; Benton et al., 2024) also explore the mixing rate and convergence bounds of diffusion models. Similarly, in optimization processes, when following the gradient vector field, trained models can converge to the same parameter distribution (Zhao et al., 2020; Cazenavette et al., 2022).

# 3 OUR METHOD

In this section, we introduce our algorithm Gradient Trajectory Pursuit (GTP), a framework designed to select influential language data through joint data selection. We start from introducing the problem formulation in section 3.1, discussing the base rationale behind trajectoy matching, then explain the main algorithm and the distribtued variants in section 3.2. Finally, we discuss the complexity and computation time of our algorithm in section 3.3.

## 3.1 PROBLEM STATEMENT

**Problem definition.** Given a large collection of training samples $\mathcal{D}_{tr} = \{(\boldsymbol{x}_i, \boldsymbol{y}_i)\}_{i=1}^N$, the goal is to select a small subset $\mathcal{D}_S = \{(x_{s_i}, y_{s_i})\}_{i=1}^M$ that maximizes the generalization performance of models trained on it when evaluated on a test set $\mathcal{D}_{te}$. This test set may either be drawn from the same distribution as the training data or from a different, task-specific distribution, leading to *in-domain* or *targeted-domain* data selection.

**Main idea.** We approach the subset selection problem from considering model training dynamics. With a training set, the gradient descent process converges to the posterior distribution $p(\boldsymbol{\theta}|\mathcal{D}) \propto p(\mathcal{D}|\boldsymbol{\theta})$ over the model parameter $\boldsymbol{\theta}$. Assuming the existence, if a selected subset $\mathcal{D}_S$ has a gradient vector field w.r.t. $p(\boldsymbol{\theta}|\mathcal{D}_S)$ that closely matches the gradient vector field of the full dataset, a model trained using the subset can converge to the same distribution over $\boldsymbol{\theta}$. Considering the intensive computation cost for computing and matching over the whole gradient vector field[1], we only focus on a part of parameter states obtained from teacher training and perform matching in a subspace. This draws inspiration from findings that the optimization process actually happens in a subspace (Gur-Ari et al., 2018; Frankle & Carbin, 2018; Singhal et al., 2023) where a random guess can already perform decently well. We use parameter states from early trajectories, assuming that the early training provides the most diversity and larger gradient subspace since optimization is an information collapse process. To minimize the matching objective, we collectively solve $|\mathcal{D}_S|$ samples, and perform joint selection among all data points. This differs from previous works, where Killamsetty et al. (2021) focuses on online settings (ours is offline) and uses OMP to select one sample per iteration (leading to scaling issue in offline cases), while compared to Xia et al. (2024) which uses

---

[1]For example, if we store gradient of a full language model over the full training set, the storage is equivalent to saving $N$ copies of language models. Every parameter state we consider will need a set of $N$ copies, leading to $N \times$ number of states stored on disk or RAM.

---

**Algorithm 1** GTP Algorithm

---

**Require:** Target dataset $\mathcal{D}_{tar}$; train dataset $\mathcal{D}_{tr}$; model parameter trajectory $\left(\boldsymbol{\theta}^{(0)},\ \boldsymbol{\theta}^{(1)},\ ...,\ \boldsymbol{\theta}^{(T)}\right)$;
    number of selected samples $M$; gradient dimension $d$; subspace dimension $d_s$; selection algorithm ALG
    // Subtract gradients
    Get gradients of data from both train and target $\mathcal{G}^{tr}=\left(\boldsymbol{g}_0^{tr},\ ...,\ \boldsymbol{g}_T^{tr}\right)$ and $\mathcal{G}^{tar}=\left(\boldsymbol{g}_0^{tar},\ ...,\ \boldsymbol{g}_T^{tar}\right)$
    // Compute target subspace
    Obtain subspace of target gradients $\mathcal{G}^{tar}$ across timesteps: $\mathcal{U}=\left(U^0, ..., U^T\right)$
    // Project gradients
    Project training data gradient to subspace as $\mathcal{G}_{\mathcal{U}}^{tr}=\left(U^0\circ\boldsymbol{g}_0^{tr}, ..., U^T\circ\boldsymbol{g}_T^{tr}\right)$
    // Compute $\boldsymbol{A}$ and $\boldsymbol{b}$
    Concatenate $\mathcal{G}_{\mathcal{U}}^{tr}$ across timesteps and obtain $\boldsymbol{A}$                ▷ $\boldsymbol{A}$ matrix has size $|\mathcal{D}_{tr}|\times Td_s$
    // $\mathcal{G}_{\mathcal{U}}^{tar}$ is $\mathcal{G}_{\mathcal{U}}^{tr}$ if in-domain
    Mean across the batch axis of $\mathcal{G}_{\mathcal{U}}^{tar}$ and get $\boldsymbol{b}$            ▷ $\boldsymbol{b}$ vector has size $Td_s$
    // Select subset $\mathcal{S}$ using matching pursuit algorithm given $\boldsymbol{A}$ and $\boldsymbol{b}$
    **if** ALG == ITERCOSAMP **then**
        $\mathcal{S}=$ITERCOSAMP$(\boldsymbol{A}, \boldsymbol{b}, M, \#\text{iters})$            ▷ Number of iterations: #iters
    **else if** ALG == DISTCOSAMP **then**
        $\mathcal{S}=$DISTCOSAMP$(\boldsymbol{A}, \boldsymbol{b}, M, \#\text{iters}, \#\text{machines})$      ▷ Number of machines: #machines
    **end if**
    Output $\mathcal{S}$                                         ▷ $|\mathcal{S}|=M$

---

independent selection, our algorithm performs joint selection to achieve better performance. We do not make specific assumptions on in-domain or target-domain and use $\mathcal{D}_{tar}$ to denote the target dataset, which can come from either in-domain data or target domains. Our full algorithm is summarized in algorithm 1. The detailed algorithm components and derivation are discussed in the following section.

### 3.2 GRADIENT TRAJECTORY PURSUIT

We denote the parameter states on the optimization trajectory as $\boldsymbol{\tau}=\left(\boldsymbol{\theta}^{(0)},\boldsymbol{\theta}^{(1)},...,\boldsymbol{\theta}^{(T-1)}\right)$. $T$ can be manually set. To select the index subset $S$ that matches the gradient of target dataset in a subspace, we define the following objective function:

$$\mathcal{L}_t(S)=\left|\left|U^t\circ\nabla_{\boldsymbol{\theta}^{(t)}}\log p(\boldsymbol{\theta}^{(t)}|\mathcal{D}_{tar})-U^t\circ\nabla_{\boldsymbol{\theta}^{(t)}}\log p(\boldsymbol{\theta}^{(t)}|\mathcal{D}_S)\right|\right|_2 \tag{1}$$

where $\mathcal{L}_t$ is the per-step matching loss, $U^t\circ\boldsymbol{g}$ denotes the projection of vector on a subspace $U^t$, $\mathcal{D}_{tar}$ is the full dataset for in-domain case and target-domain dataset otherwise. We would like to minimize the summation across steps as the final loss: $\mathcal{L}(S) = \sum_t \mathcal{L}_t(S)$. To formulate the selection set $S$ into the objective, we introduce a set of weights $w_i$ for each data point and define the gradient as $\nabla_{\boldsymbol{\theta}}\log p(\boldsymbol{\theta}|\mathcal{D}_S)=\sum_{i=1}^N w_i\nabla_{\boldsymbol{\theta}}\log p((\boldsymbol{x}_i,\boldsymbol{y}_i);\boldsymbol{\theta})$, $(\boldsymbol{x}_i,\boldsymbol{y}_i)\sim\mathcal{D}_{tr}$, where $\boldsymbol{w}=(w_1,...,w_N)$ is regularized through L0 norm $||\boldsymbol{w}||_0$. Due to sparsity enforced in $L0$ norm, the indices of final weights $\boldsymbol{w}$ that are non-negative is the final selected set $S$. The objective function with index set incorporated is then:

$$\mathcal{L}(S)=\sum_t\left|\left|U^t\circ\nabla_{\boldsymbol{\theta}^{(t)}}\log p(\boldsymbol{\theta}^{(t)}|\mathcal{D}_{tar})-\sum_{i=1}^N w_i U^t\circ\nabla_{\boldsymbol{\theta}^{(t)}}\log p((\boldsymbol{x}_i,\boldsymbol{y}_i);\boldsymbol{\theta}^{(t)})\right|\right|_2+\left|\left|\boldsymbol{w}\right|\right|_0 \tag{2}$$

Note that directly optimizing the above objective function requires a proper algorithm to minimize the non-differentiable $L0$ norm in equation 2. We will discuss the core solver variants that jointly minimize equation 2 in later this section.

**The evolving subspace.** We choose to project the model gradients onto a subspace before matching, both to reduce the storage and computation cost and to rule out potential noise signals (Singhal et al., 2023). As training progresses, parameter state at each step $t$ can have a different meaningful subspace $U^t$. We parallelize the subspace computation across and obtain a series of evolving subspace $\left(U^0, ..., U^{T-1}\right)$. To compute the subspace, we use the full training set $\mathcal{D}_{tr}$ for in-domain and use target dataset $\mathcal{D}_{target}$ for target-domain case. Note that different subspace analysis algorithms are applicable here. We adopts standard PCA for computing the principal components for the subspace.

**Algorithm 2** Iterative Compressive Sampling

1: ITERCOSAMP($\boldsymbol{A}$, $\boldsymbol{b}$, $M$, $K$)
2: // Iteratively apply compressive sampling for minimizing the residual via selecting a subset of data points.
3: // Initialize residual and empty subset.
4: $\boldsymbol{r}^0 \leftarrow \boldsymbol{b}$
5: $\mathcal{S}^0 \leftarrow \emptyset$
6: **repeat**
7:     $\boldsymbol{p} \leftarrow \text{dot}(\boldsymbol{A}, \boldsymbol{b})$      ▷ Similarities
8:     $\boldsymbol{\Omega} \leftarrow \text{supp}(\boldsymbol{p}_{|2M})$    ▷ Top $2M$ largest
9:     $\boldsymbol{T} \leftarrow \boldsymbol{\Omega} \cup \mathcal{S}^{k-1}$
10:   // Nonnegative Least Square with submatrix
11:     $\boldsymbol{w} \leftarrow \text{NNLS}(\boldsymbol{A}_{|T}, \boldsymbol{b})$
12:     $\mathcal{S}^k \leftarrow \text{supp}(\boldsymbol{w}_{|M})$    ▷ Top $M$ largest
13:     $\boldsymbol{w}' \leftarrow \text{NNLS}(\boldsymbol{A}_{|\mathcal{S}^k}, \boldsymbol{b})$
14:     $\boldsymbol{r}^k \leftarrow \boldsymbol{b} - \text{dot}(\boldsymbol{w}', \boldsymbol{A}_{|T})$
15: **until** $K$ times
16: **Return** $\mathcal{S}^K$

**Algorithm 3** Distributed Compressive Sampling

1: DISTCOSAMP($\boldsymbol{A}$, $\boldsymbol{b}$, $M$, $K$, $N$)
2: $\boldsymbol{r}^0 \leftarrow \boldsymbol{b}$
3: $\mathcal{S}^0 \leftarrow \emptyset$
4: $\boldsymbol{A} \leftarrow \text{PARTITIONCOLUMN}(\boldsymbol{A}, N)$
5: $\boldsymbol{b} \leftarrow \text{PARTITIONCOLUMN}(\boldsymbol{b}, N)$
6: // Each machine handles a col-separated $\boldsymbol{A}$,$\boldsymbol{b}$
7: **repeat**
8:     $\boldsymbol{p} \leftarrow \text{dot}(\boldsymbol{A}, \boldsymbol{b})$      ▷ Similarities
9:     $\boldsymbol{p} \leftarrow \text{GATHERSUM}(\boldsymbol{p})$
10:     $\boldsymbol{\Omega} \leftarrow \text{supp}(\boldsymbol{p}_{|2M})$    ▷ Top $2M$ largest
11:     $\boldsymbol{T} \leftarrow \boldsymbol{\Omega} \cup \mathcal{S}^{k-1}$
12:     $\boldsymbol{w} \leftarrow \text{NNLS}(\boldsymbol{A}_{|T}, \boldsymbol{b})$
13:     $\boldsymbol{w} \leftarrow \text{GATHERSUM}(\boldsymbol{w})$
14:     $\mathcal{S}^k \leftarrow \text{supp}(\boldsymbol{w}_{|M})$    ▷ Top $M$ largest
15:     $\boldsymbol{w}' \leftarrow \text{NNLS}(\boldsymbol{A}_{|\mathcal{S}^k}, \boldsymbol{b})$
16:     $\boldsymbol{r}^k \leftarrow \boldsymbol{b} - \text{dot}(\boldsymbol{w}', \boldsymbol{A}_{|T})$
17: **until** $K$ times
18: **Return** $\mathcal{S}^K$

**The non-negative least square problem.** To simplify the notations of equation 2, we concatenate the projected gradient vector $U^t \circ \nabla_{\boldsymbol{\theta}^{(t)}} \log p(\boldsymbol{\theta}^{(t)}|\mathcal{D}_{tar})$ across steps as $\boldsymbol{b} \in \mathbb{R}^{T \times D}$ where $D$ is the subspace size, and build a matrix $\boldsymbol{A}$ where each row is the projected gradient vector for a data point. With non-negative weight vector $\boldsymbol{w} \in \mathbb{R}^{1 \times N}$, we arrive at a simple non-negative least square problem with $L0$ norm as regularization:

$$\min_{\boldsymbol{w}} ||\boldsymbol{w}||_0 \quad \text{subject to } ||\boldsymbol{A}\boldsymbol{w} - \boldsymbol{b}||_2^2 = 0 \tag{3}$$

The above is the equation 2 with simplified notation. To solve the objective function, we discuss two algorithms: Iterative Compressive Sampling Pursuit and Distributed Compressive Sampling Pursuit. The two algorithms are summarized in algorithm 2 and 3.

**Iterative Compressive sampling pursuit.** Equation 3 cannot be directly minimized through differentiation. Instead, a greedy selection algorithm that solves this problem is developed in Needell & Tropp (2009) through iterative greedy selection based on a residual vector. In the selection process, $2M$ top data points are jointly selected all at once, a non-negative least square step is used to adjust the weights (i.e. implicit deduplication), then the re-ranked top $M$ samples are used to update the residual vector. Compared to orthogonal matching pursuit (OMP) used in Killamsetty et al. (2021), the algorithm is much faster and scalable.

**Distributed Compressive sampling pursuit.** Given the potential larger number of time steps in consideration, and the amount of data samples to be selected, compressive sampling matching pursuit can still be not mostly optimal in terms of computation time. We further design an algorithm that can distribute the pursuit process across multiple machines. We use a star model, where one center machine communicates with multiple machines and adopt column partition to build separate $\boldsymbol{A}$s that are distributed and parallelized. Details are shown in algorithm 3.

### 3.3 COMPUTATION TIME ANALYSIS

As pointed out in Xia et al. (2024), the computation time cost for all steps should be considered. Our algorithm consists of warmup model training, gradient subtraction and storing, subspace computation, and data selection, which follows the design in Xia et al. (2024). The main difference come from our subspace computation and selection algorithm, where we adopt PCA instead of random projection and use GTP instead of top-$k$ for data selection.

## 4 EXPERIMENTS

We evaluate our data selection algorithm on standard benchmarks against gradient-based prior arts. In section 4.1, we perform in-domain data selections on large language model agent tasks. In section 4.2, we compare algorithms on targeted instruction tuning, a benchmark focusing on empowering language model to adeptly follow human instructions.

| Method | 5% | 10% | 15% | 20% | Full data |
|---|---|---|---|---|---|
| #Samples | 500 | 1000 | 1500 | 2000 | 10000 |
| Random | 42.7 (2.8) | 61.3 (3.5) | 71.7 (3.1) | 80.5 (3.1) | |
| LESS$_{SGD}$ (top-$k$) | 38.9 (1.7) | 41.6 (1.2) | 42.4 (2.2) | 43.2 (0.9) | |
| G-DIG | 48.8 (4.2) | 58.4 (3.5) | 70.1 (1.9) | 75.6 (2.1) | |
| RDS (representation-based) | 37.3 (1.8) | 63.0 (2.1) | 74.9 (2.4) | 80.8 (1.2) | |
| **GTP - full (ours)** | **52.7 (3.0)** | **72.1 (2.8)** | **77.2 (2.4)** | **82.1 (2.9)** | 85.6 (2.6) |
| Δ (improvement) over random | **10%** | **10.8%** | **5.5%** | **1.6%** | |
| Δ (improvement) over top-$k$ | **13.8%** | **30.5%** | **34.8%** | **38.9%** | |
| **GTP - distributed (5 machines)** | **50.4 (2.9)** | **71.6 (2.8)** | **76.2 (1.9)** | **81.6 (1.7)** | |
| Δ (improvement) over random | **6.6%** | **10.3%** | **4.5%** | **0.8%** | |
| Δ (improvement) over top-$k$ | **10.4%** | **30.0%** | **33.8%** | **38.1%** | |

Table 1: Comparison across methods on ALFWorld expert demonstration dataset under various budgets. Top-$k$ selection leads to data points that are more similar (duplicate in terms of information). Joint selection using GTP outperforms all baseline methods consistently.

## 4.1 LLM AGENT: ALFWORLD

**Benchmark setup.** ALFWorld is a suite of text environments built upon interactive Textworld (Côté et al., 2019) that benchmarks agents on solving sequential tasks. Typical tasks include giving agents the current partial observed environment information and require the agents to achieve a goal (e.g., finding objects).

We focus on offline imitation learning settings. For train and test datasets, we use offline trajectories from the official ALFWorld repository (Shridhar et al., 2021) which consists of 3,553 tasks, each with an average of 19 steps. The training set contains $48k$ state-action pairs, and test set totals at 1,861 state-action pairs. In practice, we find that using $10k$ training data already achieves similar generalization performance compared to full training set, i.e., dataset reduction beyond $10k$ is not very meaningful since the selection can be easily done through random subset picking, and thus choose to use a subset with $10k$ data points as our main subject of study. In model evaluation, for convenience, we directly compare the model's per-step action prediction accuracy. Per-step accuracy is known to be reflective on model's ability and has a strong correlation with final success rates.

**Model setup.** For evaluating whether a dataset is effective for training, we use supervised fine-tuning on OPT-125M (Zhang et al., 2022) with LoRA (Hu et al.) as the basic training setup. Given a selected subset dataset for training, all models are optimized using Adam (Kingma & Ba, 2014) with a learning rate 5e-4 and batch size 80 for 20 epochs to ensure its convergence. We follow the most recent prior method (Xia et al., 2024) to generate a warm-up training trajectory for gradient extraction and data selection. The warm-up trajectory is trained using learning rate 1e-4 and batch size 10 for 5 epochs. A model checkpoint is saved every 500 interations, resulting in model states on across timesteps.

**Baselines.** The most relevant prior method is **LESS** Xia et al. (2024), where we build upon most of its pipeline, including warm-up training and gradient computation. LESS uses top-$k$ selection with scores maxing across pre-defined subtasks, which does not exist in ALFWorld. We instead directly compare individual gradient vectors with gradient trajectories. **G-DIG** (Pan et al., 2024) is a baseline that uses clustering directly on gradients and samples within each cluster. Following (Xia et al., 2024), we also compare with **RDS** (Representation-based Data Selection) Zhang et al. (2018); Hanawa et al. (2020), which uses the mean of last layer hidden states for selection.

**Main results.** We summarize the main results in table 1. As demonstrated in previous works (Xie et al., 2023; Xia et al., 2024; Pan et al., 2024), random sampling is a strong baseline to beat. Gradient-based algorithms tend to pick highly redundant examples when matching gradients. LLM Agent benchmarks contain traces with incremental steps in the training data, and need algorithms to be sensitive to sample duplication. *Why does top-k fail?* We visualize the selected top three examples based on the scores computed from two algorithm in figure 2, showing that the initial top k selections are highly duplicated, while after our pursuit algorithm the selected samples exhibit more diversity. Representation-based selection (RDS) shows higher performance since it uses the features from pretrained model. Across different data ratios, our algorithm, both standard and distributed versions, consistently outperform the baselines. For the distributed version, we use separate the 10 timesteps into 5 machines.

**Top-K**

★ [GOAL] put a remote control in armchair. [STATE 0] You are in the middle of a room. Looking quickly around you, you see a armchair 1, a armchair 2, a armchair 3, a coffee table 1, a dining table 1, a dining table 2, a garbage can 1, a sofa 1, and a tv stand 1…

★ [GOAL] put a remote control in armchair. [STATE 0] You are in the middle of a room. Looking quickly around you, you see a armchair 1, a armchair 2, a armchair 3, a coffee table 1, a dining table 1, a dining table 2, a garbage can 1, a sofa 1, and a tv stand 1…

★ [GOAL] put a remote control in armchair. [STATE 0] You are in the middle of a room. Looking quickly around you, you see a armchair 1, a armchair 2, a armchair 3, a coffee table 1, a dining table 1, a dining table 2, a garbage can 1, a sofa 1, and a tv stand 1…

**GTP**

★ [GOAL] look at alarmclock under the desklamp. [STATE 0] You are in the middle of a room. Looking quickly around you, you see a bed 1, a desk 1, a drawer 1, a drawer 2, a drawer 3, a drawer 4, a drawer 5, a dresser 1, a garbage can 1, a shelf 1, a shelf 10, a shelf 11, …

★ [GOAL] put some remote control on armchair. [STATE 0] You are in the middle of a room. Looking quickly around you, you see a armchair 1, a coffee table 1, a drawer 1, a drawer 2, a drawer 3, a drawer 4, a drawer 5, a drawer 6, a drawer 7, a garbage can 1, a side table 1, a side table 2, a side table 3, …

★ [GOAL] put a remote control in armchair. [STATE 0] You are in the middle of a room. Looking quickly around you, you see a armchair 1, a armchair 2, a armchair 3, a coffee table 1, a dining table 1, a dining table 2, a garbage can 1, a sofa 1, and a tv stand 1…

Figure 2: Three examples with topmost scores from selection algorithms. *Left:* Top-$k$ tends to select redundant samples due to lack of de-duplication mechanisms. *Right:* GTP instead chooses diverse examples.

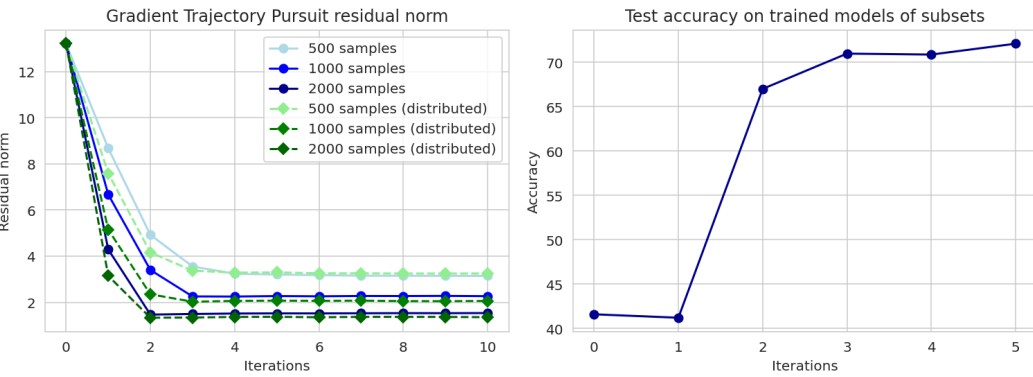

Figure 3: Residual norms across iterations.      Figure 4: Subset accuracy across iterations.

**Residual norms across iterations.** We rollout 10 steps of compressive sampling matching pursuit iterations and monitor the L2 norm of residual vectors. The residual norm trends are shown in figure 3. Typically the algorithm converges after 5 steps.

**Correlation between residual norm and model performance over iterations.** The initial step of GTP is equivalent to a top-$k$ selection, and later iterations of GTP algorithm can be seen as a process of de-duplication through adjusting the data weights $w$. In figure 4, we plot the accuracies of subsets of 1000 samples selected after each GTP iteration. Through progressively solving the data weights $w$, the performance gradually improves. We only plot 5 steps since the algorithm plateaus (see figure 3).

**How does the algorithm computation time scale with number of samples?** As another algorithm for joint data selection, orthogonal matching pursuit (OMP) can also collectively solve data weights for de-duplication (Killamsetty et al., 2021). We compare our algorithm GTP, its distributed version, and OMP across various number of budget samples. OMP selects only one sample per iteration while each requires a separate solving of non-negative least squre problem. This rapidly increases the computation time with more samples. Our algorithms, based on compressive sampling process, can scale well and have a much more mild computation time curve.

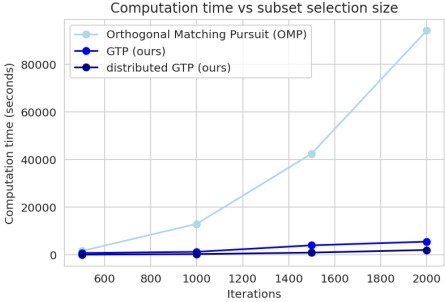

## 4.2 TARGETED INSTRUCTION TUNING

**Task setup.** We follow the setup in Xia et al. (2024) and evaluate our algorithm on the *targeted instruction tuning* task. Compared to standard instruction tuning where mixed datasets are used to finetune large language model to follow human instructions, targeted instruction tuning individually selects data samples for different target benchmarks for finetuning language models. The selection algorithm hence plays an important role for improving the final trained model performance. For example, multiple works find that when various data sources are mixed together, it can negatively impact the model performance for different target tasks Wang et al. (2023a); Xia et al. (2024), while Xia et al. (2024) achieves equivalent or better results with only 5% of the full dataset.

**Datasets and models.** We follow the settings from Xia et al. (2024) and use a diverse and mixed collection of public instruction tuning datasets as the main training set to select from. The datasets include: Flan V2 (Longpre et al., 2023), CoT (Wei et al., 2022), Dolly (Conover et al., 2023), and Open Assistant (Köpf et al., 2024). The four datasets are commonly used in various models and benchmarks. For the language model being tested upon, we use Mistral-7B (Jiang et al., 2023)[2].

**Implementation details.** To build the initial parameter states for extracting gradients, we warmup train Mistral-7B (also with LoRA) for 4 epochs using 5% of the full dataset. Data gradient extractions are performed in parallel across machines. We use the selected data from each benchmark as the target dataset and compute subspace and target gradient $b$. These are exactly following the standard setups in Xia et al. (2024). The GTP iterations are performed 5 times to select the final subset.

**Evaluation benchmarks and results.** There are three targeted benchmarks considered in the task: MMLU (Hendrycks et al., 2020), BBH (bench authors, 2023), and TydiQA (Clark et al., 2020). MMLU is a *de-facto* benchmark for evaluating large language model's capability on 57 subsets across elementary mathematics, humanities, law, social sciences and more. BBH evaluates the reasoning capability of large language models. It is curated as a subset of the BIG-Bench benchmark, focusing on 27 tasks where current models struggle to match human performance. TyDiQA is a multilingual question-answering benchmark covering 11 diverse languages, and features questions from

|  | MMLU | BBH | TydiQA |
|---|---|---|---|
| Base | 62.4 | 57.7 | 52.2 |
| Rand | 60.0 | 54.5 | 56.9 |
| LESS | 61.8 | 56.0 | 60.3 |
| **GTP (ours)** | **62.0** | **60.0** | **64.5** |
| $\Delta$ **(improv.)** | **+0.2** | **+4.0** | **+4.2** |
| #Samples | 1353 | 1353 | 1353 |
| Runtime (seconds) | 800 | 355 | 65 |

Table 2: Comparison of our algorithm with state-of-the-art algorithm (Xia et al., 2024).

native speakers seekingn answers. *Main result.* The test accuracies of models trained on MMLU, BBH, and TydiQA benchmarks are summarized in table 2. We observe that our algorithm outperforms the prior method (Xia et al., 2024) by 0.2% on MMLU, 4.0% on BBH, and 4.2% on TydiQA using only 0.5% of the full dataset, achieving a 10x reduction on the selected subset size and achieves better results.

## 5 CONCLUSIONS

In this paper, we propose an algorithm (GTP) that can both perform joint data selection and has the scalability towards larger target sample sizes. The algorithm automatically de-duplicates samples and can potentially serve as a standard selection technique other than top-$k$. Both in-domain selection and target-domain selection are applicable for GTP. For potential limitations, gradient-based methods still rely on a few steps of model training, which can be more expensive than pure representation-based methods or influence function methods. It will be worth attempting to connect matching pursuit with representation-based or influence function methods and develop an algorithm that does not require additional model training for data selection.

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
