# OpenReview forum: "Influential Language Data Selection via Gradient Trajectory Pursuit"
_ICLR.cc/2025/Conference — ICLR 2025 Conference Withdrawn Submission_

### Official Review · Reviewer_KHGh · 2024-10-16

**Soundness:** 3
**Presentation:** 2
**Contribution:** 3
**Rating:** 6
**Confidence:** 2

**Summary:**

The paper proposes a technique to select influential subsets of language data based on a subset budget. The ideas is based on matching trajectories in a subspace, and is claimed to be 17x more efficient compared to vanilla matching pursuit. For targeted instruction-tuning tasks, whole-dataset performance can be achieved using only 0.5% of the data.

**Strengths:**

-- This type of dataset selection technique is generally important and timely.

-- The paper is clearly written (despite various small types), and has enough detail that what's described should be more-or-less reproducible.

-- The results appear fairly impressive, though seem to be fairly modest compared to the strongest baseline

-- Reasonably strong methodology paper, though the idea is ultimately fairly simple and described in pretty clear terms.

**Weaknesses:**

-- The contribution compared to LESS seems not so large, and the performance improvements are also fairly incremental (though they are still significant)

-- Experiments seem overall somewhat less thorough (e.g. in terms of models being compared) than in prior work (such as LESS), though the authors do have some explanations for this

-- Hard to totally make sense of some of the experimental results in a few places

**Questions:**

-- Can you explain more about why Llama couldn't be compared against? I am not aware of the issue described

-- Are all the methods in Table 2 subject to the same subset size constraint? I assume they are, but it's not totally clear from the wording of the results section, which seems to imply that smaller subsets are chosen

---

### Official Review · Reviewer_y4B9 · 2024-11-02

**Soundness:** 2
**Presentation:** 2
**Contribution:** 2
**Rating:** 5
**Confidence:** 3

**Summary:**

**Summary:**

In this paper, the authors extend the work of Xia et al. (2024) to subset selection settings, where the selection mechanism relies on the collaborative effect of chosen samples rather than a top-k selection paradigm. Experimental results are provided to showcase the effectiveness of the proposed method.

**Strengths:**

**Strengths:**

Incorporating the combinatorial effect into data selection is intuitive, as a top-k selection approach may not fully capture the collaborative potential among data samples.

The method demonstrates considerable performance improvement over the vanilla LESS method on the ALFWorld dataset.

**Weaknesses:**

**Weaknesses:**

As noted in subsection 3.3, the main distinction from Xia et al. (2024) lies in the departure from the top-k selection paradigm by incorporating ideas from Needell & Tropp (2009), while most other components remain unchanged.

The computational time analysis in subsection 3.3 is unclear. There is no analysis of computational complexity, nor are there empirical runtime results or GPU memory cost reports. Additionally, PCA may present a computational challenge in large-scale experiments.

The experimental scope is limited. Although multiple datasets are utilized, the main experiment comparing multiple baselines is only conducted on the ALFWorld dataset, with only LESS compared across three instruction-tuning datasets. Specifically, it would be beneficial to see a comparison of performance and computational costs with the Random baseline and G-DIG. Furthermore, the authors could explain why the naïve random selection strategy tends to outperform most baselines in this experimental setup.

**Questions:**

Please see the weakness points above.

---

### Official Review · Reviewer_2nyj · 2024-11-04

**Soundness:** 3
**Presentation:** 2
**Contribution:** 3
**Rating:** 5
**Confidence:** 3

**Summary:**

The paper introduces Gradient Trajectory Pursuit (GTP), which jointly selects data samples based on gradient trajectory matching, optimizing over an L0-norm regularized objective. It avoids data duplication by considering the entire set of selected data points rather than independently selecting the top-k samples. The algorithm uses compressive sampling techniques, significantly improving efficacy, and further supports a distributed framework for handling large datasets. Empirical results on two benchmark datasets demonstrate consistent performance of GTP over existing top-k selection and other gradient-based baselines.

**Strengths:**

S1: With the increasing model and dataset size, it is an important field to study effective data selection for LLMs. And the contribution of this paper seems non-trivial to the field.

S2: The overall idea of performing joint selection based on gradient trajectory matching is intuitive and reasonable.

S3: The empirical results based on ALFWorld and three instruction tuning evaluation sets are consistently better than the considered baselines.

**Weaknesses:**

W1: The writing and presentation of the paper need substantial improvement. It is very unclear how Equation 1 is derived from the main idea discussed in Section 3.1. The three algorithm blocks should also be described with further details in the content sections. In general, I find Section 3 very hard to follow.

W2: The scope of the current experiments seems limited. It would be great to experiment with multiple LLMs from different model families and sizes for each baseline.

W3: It would be great to include some qualitative analysis based on the 5% examples selected by your method. The findings there could be very valuable.

**Questions:**

Q1: How do you obtain $\nabla_{\mathbf{\theta}^{(t)}} \log p(\mathbf{\theta}^{(t)} | \mathcal{D}_\text{tar})$ in Equation 1?

Q2: How do you obtain $\nabla_{\mathbf{\theta}^{(t)}} \log p((\mathbf{x}_i, \mathbf{y}_i); \mathbf{\theta}^{(t)})$ in Equation 2?

Q3: What exactly is the result of your computation time analysis in Section 3.3?

Q4: how do you obtain the model parameter trajectory as indicated in your Algorithm 1?

Q5: What exactly do "supp" and "NNLS" mean in Algorithm 2?

---

### Official Review · Reviewer_uXZc · 2024-11-04

**Soundness:** 2
**Presentation:** 3
**Contribution:** 2
**Rating:** 3
**Confidence:** 5

**Summary:**

The authors of this paper studied data selection problem for large language models. The basic idea is to select a subset of datapoints from a larger training set, where the gradients of the objective function with respect to a given set of selected model parameters match those computed from the complete dataset. To reduce computation complexity, the matching is performed in the projected gradient space and experiment results were reported on the ALFWorld dataset.

**Strengths:**

1.	Data selection for effective model training has become a very important problem for large language models. Hence, this submission touches upon a timely issue.
2.	The descriptions of the proposed solution and related work are largely clear and easy to follow. This helps readers to understand the essence of this submission.

**Weaknesses:**

1.	It is well known that modern large language model training has several stages, e.g., pre-training and post-training. Given the submission is targeted at language model training, it is very unclear which stage the proposed method is designed for. Since different stages have distinct purposes, specific designs are needed, though their loss function might look similar (e.g., next token prediction).
2.	The proposed solution is very similar to the frequently mentioned LESS algorithm (Xia et al. 2024) in the paper, and this submission just changes their cosine metric to L2 norm, with an additional sparsity regularization. And of course, one advantage of the proposed solution over LESS is it can use some less greedy solution to choose data points, which seem indeed brings performance gain to LESS.
3.	The proposed solution requires to match gradients on a set of given model parameters. It is very unclear in practice how to most effectively select this set of parameters to compute the gradient. The manuscript gives a very heuristic approach: using the model parameters obtained from early stage of training. This comes to my first question: which stage of LLM training it is. The sentence alone seems suggest pre-training (i.e., “a random guess can already perform decently”, “early trajectories”), but the experiment uses pretrained model, OPT-125M, which clearly is not a random guess.
4.	It is very unclear why the proposed solution can really solve the complicated data selection problem for LLMs. As the authors mentioned, matching/controlling the training dynamics of LLMs is important, which govern where the final model will end up. But the proposed solution did not touch this aspect at all, e.g., the optimizer and learning rate schedule. Matching gradients on some selected points cannot suggest anything about the model training dynamics on the selected subset. And therefore, it is unclear why the selected data points can bring any good properties to the trained model.

**Questions:**

1.	How do we select the set of model parameters to compute the gradients? And how large this set should be?
2.	Will the selected subset be robust to the later choices of model training, e.g., optimizer configuration, learning rate schedule? If so, why is that?
3.	Will the proposed solution readily applicable to different stages of LLM training?

---

### Note · Authors · 2024-11-21

I have read and agree with the venue's withdrawal policy on behalf of myself and my co-authors.